# Local Delivery Strategies for Peptides and Proteins into the CNS: Status Quo, Challenges, and Future Perspectives

**DOI:** 10.3390/ph16060810

**Published:** 2023-05-30

**Authors:** Weizhou Yue, Jie Shen

**Affiliations:** 1Department of Biomedical and Pharmaceutical Sciences, University of Rhode Island, Kingston, RI 02881, USA; weizhou_yue@uri.edu; 2Department of Chemical Engineering, University of Rhode Island, Kingston, RI 02881, USA

**Keywords:** peptide/protein delivery, local therapeutic delivery, central nervous systems, nanotechnology-based delivery systems

## Abstract

Over the past decades, peptides and proteins have been increasingly important in the treatment of various human diseases and conditions owing to their specificity, potency, and minimized off-target toxicity. However, the existence of the practically impermeable blood brain barrier (BBB) limits the entry of macromolecular therapeutics into the central nervous systems (CNS). Consequently, clinical translation of peptide/protein therapeutics for the treatment of CNS diseases has been limited. Over the past decades, developing effective delivery strategies for peptides and proteins has gained extensive attention, in particular with localized delivery strategies, due to the fact that they are capable of circumventing the physiological barrier to directly introduce macromolecular therapeutics into the CNS to improve therapeutic effects and reduce systemic side effects. Here, we discuss various local administration and formulation strategies that have shown successes in the treatment of CNS diseases using peptide/protein therapeutics. Lastly, we discuss challenges and future perspectives of these approaches.

## 1. Introduction

Proteins are typically comprised of 50 or more amino acids, whereas peptides contain 2–50 amino acids, and both are essential nutrients for cells and some of the most primary molecules involved in living creatures. The term “protein”, originating from the Ancient Greek word πρῶτος (prôtos, “first”), first appeared in the literature in 1838 in a letter from Swedish chemist Jöns Jacob Berzelius to Dutch chemist Gerardus Johannes Mulder [1]. With the advances in the field of biomedical sciences, synthetic peptides/proteins and recombinant protein technology have been commercially available since 1955. Recombinant human insulin has been the most important recombinant protein therapeutic for diabetes in the past few decades [2]. Owing to their high potency, specificity, and various biological functions involved in the progression and/or inhibition of human diseases, peptide and protein therapeutics have gained extensive attention.

Peptides and proteins are important signals or languages of neurological diseases and disorders such as lysosomal storage diseases (LSDs), Alzheimer’s disease (AD), Parkinson’s disease (PD), amyotrophic lateral sclerosis (ALS), and brain cancer. They execute various biological functions in the brain, such as neurotransmission and neuromodulation, hormonal regulation in the endocrine system, regulation of the immune system, the cerebral blood flow, cerebrospinal fluid (CSF) secretion, and the internal environment of the brain (e.g., water and electrolyte content), and they also modulate the blood brain barrier’s (BBB) permeability to nutrients [3]. Abnormal expression of signal peptides and/or proteins has been shown to be closely associated with disease status [4]. For example, low levels of vascular endothelial growth factor (VEGF) can lead to motor neuron degeneration [5]. Inadequate insulin secretion may increase the risk of developing AD [6]. Moreover, a low neuropeptide Y level in CSF correlates with the intrusive memories of trauma, a symptom of post-traumatic stress disorder (PTSD) [7]. Therefore, peptides and proteins are good therapeutic candidates for neurological diseases.

The brain is a soft tissue that is uniquely protected by internal and external physiological barriers such as the skull and vertebral column, meninges, the blood-CSF barrier, and the BBB [8]. Over 98% of small molecular therapeutics are unable to enter the brain through systemic administration due to the presence of the BBB [9], which possesses even greater challenges for proteins and peptides that are hydrophilic and with large sizes to enter the brain. There are other challenges associated with peptides and proteins for the treatment of CNS diseases, including but not limited to: (1) Inherent instability issues since their primary structure maintained by peptide bonds and the complex three-dimensional (3D) structure that is essential for their functionality are sensitive to chemical and/or enzymatic degradation in the physiological environment [10]; (2) Relatively large molecular sizes make them possess poor ability to cross physiological barriers such as the BBB, and hence, have poor biodistribution in the CNS sites of action [10]; and (3) A short plasma half-life associated with fast renal clearance and inherent instability [11]. Over the past decades, various strategies such as intra-arterial administration and local drug delivery strategies have shown successes in delivering peptides and proteins into the brain to achieve therapeutic effects [12].

Local drug delivery strategies (Figure 1), such as intracerebroventricular (ICV), intraparenchymal convection-enhanced (CED), and intrathecal (IT) administration, directly introduce therapeutics to different regions in the CNS, including the ventricles, the brain parenchyma, and the subarachnoid space of the spinal cord, which can not only bypass the BBB but also minimize systemic drug exposure and subsequent systemic side effects and toxicity [13]. On the other hand, advanced drug delivery systems, such as nanoparticles or long-acting implants or injectables, have been used to improve transport and/or prolong the release duration of macromolecular therapeutics in the CNS [14]. These advanced drug delivery systems can augment therapeutic effects of macromolecular therapeutics when administrated locally in the CNS. This review will discuss the status quo and future perspectives on local delivery of peptide and protein drugs into the CNS.

## 2. Local CNS Administration Routes and Challenges

The main approaches for delivering peptide and protein therapeutics into the CNS are through local CNS administration routes (e.g., ICV and intraparenchymal CED injection) since they can avoid the harsh environment in the gastrointestinal tract compared to oral administration and achieve faster and more distribution of therapeutics to a target site in the brain compared to intravenous administration. IT can introduce therapeutics into either the brain or spinal cord and is less invasive and more commonly used for spinal cord diseases or conditions compared to ICV and CED. Intranasal (IN) delivery is another administration route that may achieve brain drug delivery through circumventing the BBB, peripheral metabolism, and serum clearance, and is more patient-friendly compared to other invasive local delivery strategies [15]. Table 1 summarizes representative local CNS administration routes.

### 2.1. ICV

The ICV route is one of the most commonly used CNS local drug delivery strategies to directly introduce therapeutics into the CSF in the lateral ventricle of the brain. It has been demonstrated that nearly 100% of a small molecular compound administrated was near the lateral ventricle following ICV [24]. This local strategy has been used clinically for peptide and protein therapeutics. ICV administration of Brineura (cerliponase alfa) was approved by the FDA in 2017 as an enzyme replacement therapy (ERT) for the treatment of LSDs in the CNS [16], a disease that affects about one in every 6000 live births worldwide [25,26]. In clinical studies, Schulz et al., demonstrated a slower decline rate of motor and language functions in children with tripeptidyl peptidase 1 (TPP1) deficiency disease following ICV treatment of a recombinant proenzyme form of TPP1-Brineura every two weeks at a dose of 300 mg compared to controls [17]. Seo et al., suggested that developmental intellectual decline in patients with neuronopathic mucopolysaccharidosis II was effectively prevented and stabilized through ICV administration of idursulfase-β and that the treatment had good patient tolerance [19]. ALS is a neurodegenerative disorder that majorly affects the motor system [27]. In a clinical study, Van Damme et al., showed that a detectable CSF level of VEGF in patients with ALS was achieved through daily ICV dosing of VEGF (2 mg/day), and the treatment was well tolerated in most patients [20]. The most serious adverse effect observed in this clinical study was a pulmonary embolus [20]. It has been shown clinically that the survival time of ALS patients is negatively correlated with a reduced CSF concentration of Cystatin C (CysC) [28], a major protein component of Bunina bodies that were found in the remaining healthy motor neurons of sporadic ALS [29]. In a preclinical animal study, Watanabe et al., observed the extended lifespan of ALS mice following ICV administration of recombinant human CysC, which created detergent-insoluble tetramers to form protein inclusions and preserve motor neurons [30].

Despite its effectiveness in introducing therapeutics directly into the CSF in the lateral ventricle, ICV may create a bulk flow of CSF carrying infused peptides/proteins through the ventricles and reaching the subarachnoid space where the major arteries are, thus resulting in fast clearance and systemic exposure of the peptides/proteins [31]. Noguchi et al., have shown that immunoglobulin G (IgG) following ICV administration was detectable in plasma from the CSF within 24 h [32]. As a result of the fast clearance, frequent dosing may be required, which may affect patient tolerance and compliance. Furthermore, drug penetration from CSF into the brain parenchyma is limited, particularly for macromolecules like proteins and peptides [33]. ICV delivery may potentially be associated with complications related to ICV devices (e.g., catheter implants). For example, a grade 3 infection, leakage, and an increased count of white cells in the CSF were previously reported in patients [17]. These disadvantages have limited the clinical applications of ICV.

### 2.2. CED

Intraparenchymal administration, in particular CED, can increase the local therapeutic concentration through a gradient pressure to transport a bulk flow of the therapeutic within the brain parenchyma [34]. Bobo et al., found that transferrin (Tf) distributed in the white matter immediately after CED with increased infiltration into the gray matter within one day and maintaining an appropriate infusion rate was critical for successfully achieving macromolecule distribution in the brain parenchyma [35]. In a clinical safety study, Weber et al., demonstrated that a single CED administration of interleukin 4-pseudomonas toxin (NBI-300l) was well tolerated in patients with recurrent malignant glioma in different dose groups. The positive outcome observed in this study with CED treatment alone (8.2-month median survival) as compared to patients who underwent surgical resection alone (5.8-month median survival) ensured further clinical trials of NBI-3001 via CED [36]. In a phase III clinical trial, interleukin 13-pseudomonas toxin administered via CED showed similar survival in adult glioblastoma (GBM) patients at first recurrence compared to the Gliadel^®^ wafer, the only long-acting GBM treatment approved by the U.S. FDA [37]. In addition, Bander et al., studied omburtamab, a monoclonal antibody (mAb) that binds to the surface marker B7 homolog 3 of diffuse intrinsic pontine glioma (DIPG), a pediatric brainstem tumor. The pediatric DIPG patients showed good tolerance with repeated CED infusion of omburtamab [21]. The overall median survival was 17.5 months, which exceeded the outcomes of most control groups in similar clinical studies [22].

Although CED can improve drug distribution in the brain parenchyma, some drawbacks of CED delivery of peptides and proteins into the CNS have been reported, including: (1) The low diffusion in the parenchyma remains an obstacle for some macromolecules such as glial cell line-derived neurotrophic factor (GDNF), and hence, limited therapeutic effect [38,39]; and (2) Complications arising from the invasive surgical regimen have yet to be fully studied [15,40].

### 2.3. IT

The first IT administration was performed in 1885 when James Leonard Corning, a neurologist, injected cocaine in the lower lumbar vertebrae of a dog for local spinal anesthesia purposes [41]. Later, Iliff et al., elucidated how peptides/proteins transport from the IT injection site to the brain parenchyma, which may involve the following steps [31]: (1) Pulsatile remixing of interstitial fluid (ISF) and CSF leads to the initial diffusion and distribution of therapeutics in the CSF; (2) The paravascular pathway creates the space to allow CSF and a majority of macromolecules to flow; and (3) The remaining macromolecules in the paravascular pathway are transferred to the perivascular space and brain parenchyma through pulsation-assisted translocation. IT administration of peptide/protein therapeutics has been used to treat neurodegenerative diseases such as ALS. Cudkowicz et al., demonstrated that the cerebrospinal level of recombinant human superoxide dismutase (rhSODl) was increased nearly 40-fold with daily IT infusion in ALS patients [42]. Dali et al., found that IT administration of recombinant human arylsulfatase A was well tolerated in patients and slowed down the decline in motor function over time in metachromatic leukodystrophy patients [23].

The access to the perivascular spaces through IT administration may be dependent upon the size of therapeutics, as demonstrated by Pizzo et al., through a study comparing a single-domain antibody to a full-length IgG. Moreover, co-infusion of hyperosmolar mannitol was able to improve perivascular access and distribution of the full-length IgG due to the osmosis effect [43]. However, the implantable port device designed to provide repeated access for IT delivery needs to be imbedded into patients over a long period of duration. Serious adverse events such as blood and lymphatic system disorders related to IT device failures have been reported [23].

### 2.4. Others

IN is another administration route that can be used to achieve effective CNS delivery of peptides and proteins without the need for invasive procedures. Physicochemical properties of peptides and proteins, such as molecular weight and lipophilicity, partially determine the permeation of therapeutics through the nasal epithelium barrier and subsequent in vivo fates through IN administration [44]. Falcone et al., obtained successful albumin brain delivery via IN administration and found that albumin was transported from the nasal epithelium along the olfactory nerves into the CSF, followed by diffusion into the arterial perivascular spaces and rapid distribution throughout the brain [45]. IN delivery of insulin has shown improvement of cognition and daily living activities, as well as glucose metabolism in the human AD brains [46]. In clinical studies, Craft et al., showed the efficacy of a long-acting insulin detemir and regular insulin against AD through long-term daily IN administration, and the prolonged effect on memory improvement was observed [6]. This was consistent with a previous study in which substantial memory benefits from IN delivery of rapid-acting insulin and regular insulin were acquired in normal adults [47]. Until now, various peptide and protein therapeutics have been studied for the treatment of CNS diseases via IN administration using animal models, including insulin-like growth factor-I [48], glucagon-like peptide-1 antagonist exendin-(9–39) [49], galanin-like peptide [50], interferon beta [51], and orexin A [52]. The high CNS concentration of these peptides and proteins within the first hour provided evidence of their rapid transportation into the CNS. Recently, Roltsch Hellard et al., found that IN delivery of synthetic melanocortin-4 receptors antagonist protein HS014 could mediate alcohol drinking in alcohol-dependent rats and had the same effect as ICV delivery on HS014 [53].

IN administration requires the application of therapeutics at a specific location inside the nose-nasal cribriform plate in order to effectively bypass the BBB with minimum systemic exposure [15]. It has been reported that some devices failed to accurately apply the same dose of insulin to both nostrils, which may result in insufficient and/or inconsistent drug amounts in the brain, thus compromising therapeutic effects [54]. Therefore, the device design is critical to achieve brain delivery of peptides and proteins with consistent and well-controlled doses via IN administration.

## 3. Formulation Strategies to Improve CNS Protein/Peptide Delivery

Even with the advances in local CNS delivery strategies, some peptides and proteins may still face translational challenges, such as moderate half-life in the CNS, low distribution at the target tissue, and severe side effects (e.g., immunogenicity of non-modified polypeptides) [38,39,55,56]. Formulation strategies such as peptide/protein functionalization, nanotechnology, and long-acting approaches can either achieve target delivery of therapeutics to the specific neuron populations or slow down peptide/protein clearance in the target tissues, resulting in improved therapeutic effects. Recently, the incorporation of therapeutic peptides/proteins in extracellular vesicles has been shown to decrease peptide/protein immunogenicity and improve peptide/protein stability and circulation time [57]. Another new strategy involves expressing therapeutics in live cells that can then serve as depots for prolonged therapeutic action [58]. Here, we highlight recent advances in formulation strategies and discuss the clinical applications and translational potential of the collective knowledge on advanced formulation and localized delivery strategies.

### 3.1. Peptide/Protein Functionalization Strategies

Functionalization of peptides and proteins through conjugation with functional groups such as targeting ligands (e.g., human transferrin (Tf), tetanus toxin (TTC) fragment) and polyethylene glycol (PEG) has been utilized to improve protein/peptide transport to the target site and/or in vivo half-life to exert the intended functions.

Conjugating ligands to brain cell receptors and transporters (e.g., low-density lipoprotein, insulin, and growth factor) can not only improve the stability but also achieve CNS delivery of proteins and peptides [3]. In clinical studies, Laske et al., discovered that a large recombinant toxin, Tf-CRM107, a conjugate of Tf and a genetic mutant of diphtheria toxin (CRM107), was successfully delivered into a malignant brain tumor and penetrated into the nearby brain tissue via CED. Notably, over 60% of patients with malignant gliomas had at least a 50% reduction in tumor volume following the treatment of Tf-CRM107 due to the active targeting ability of Tf [59], indicating the clinical effectiveness of CED administration of functionalized proteins/peptides. Since recombinant toxins are hybrid cytotoxic proteins that are not mutagens and do not induce chemo-resistance [60] and the production of the recombinant toxins can be easily scaled up in bacteria as homogenous proteins [61], this treatment has shown a huge potential for the treatment of malignant brain cancers. Tong et al., conjugated guanidinoneomycin, a molecular ligand with a cell-surface transporter-heparan sulfate proteoglycans, with iduronidase to improve the enzyme stability and uptake into neurons and astrocytes. IN administration of this conjugate enhanced glycosaminoglycan clearance in enzyme-deficient mice, a model of mucopolysaccharidosis IIIA [62].

Another widely studied targeting ligand was non-toxic C-fragment of TTC, a protein produced by Clostridium tetani, since its structural conformation changes in response to pH changes, leading to TTC axonal transport from motor neurons of the neuromuscular junction (NMJ) to the motor neurons of the spinal cord and brainstem (Figure 2) [63,64]. The fragment can be conjugated onto polypeptides by a periodate procedure [65], or the plasmid of fusion polypeptides can be constructed and expressed in *E. coli* [66]. The unique neuronal transport property of TTC has been utilized to medicate CNS delivery of proteins and peptides. Fishman et al., showed that intramuscular-administered TTC-conjugated horseradish peroxidase can bypass the BBB to enter the CNS in mice [65]. In contrast, no SOD1 protein was detected in the mouse brainstem and spinal cord even when an amount of SOD1 that was 30 times higher was injected intramuscularly alone [66]. Figueiredo et al., demonstrated that TTC-conjugated SOD1 underwent retrograde axonal transport and trans-synaptic transfer as efficiently as TTC alone. In a preclinical study, Chian et al., discovered that the level of exogenous hIGF-1 in the spinal cord was increased after peripheral or IT administration of TTC-conjugated hIGF-1, and this treatment prevented muscle force decline with age in ALS mice [67]. Despite its success in transporting proteins and peptides into the CNS, anti-TTC antibodies generated from tetanus vaccinations may block and remove TTC and its conjugated macromolecules before they bind to the NMJ [68].

PEGylation of macro-therapeutics is commonly used to improve in vivo stability, half-life, and reduced immunogenicity [69]. The first PEGylated CNS polypeptide drug via parenteral administration was pegaspargase (Oncaspar), which prevents CNS acute lymphoblastic leukemia or relapse by clearing leukemic cells within the brain. Kang et al., discovered that IT administration of PEG-modified fibroblast growth factor 2 (FGF2) had remarkable potential for the treatment of spinal cord injuries. Due to the protective effect of PEG, PEG-FGF2 penetrated in the injured spinal cord in rats [55].

### 3.2. Long-Acting Formulations

Long-acting dosage forms are one of the most effective strategies for localized peptide/protein delivery in the CNS. These dosage forms, including pre-formed polymeric matrices (such as films, disks, rods, or wafers) or gels (either pre-formed or in situ forming), have been developed to locally deliver peptide/protein therapeutics over extended periods of time via implantation or injection in the brain parenchyma (Table 2). Sustained peptide/protein release in the surrounding brain tissue is mainly mediated by polymer degradation and polypeptide diffusion [14]. The Gliadel^®^ wafer, a biodegradable polifeprosan 20 co-polymer matrix loaded with a chemotherapeutic carmustine (BCNU), was approved by the FDA for the treatment of newly diagnosed and recurrent GBM in 1996. A total of eight wafers can slowly release the drug over a week [70]. In addition, a n-butylidenephthalide wafer with promising preclinical results [71] is currently being studying in patients with recurrent GBM [72]. Recently, Zhang et al., used an in situ gelling formulation to deliver a protein chemotactic CXC chemokine ligand 10 (CXCL10) over 12 days, which rebuilt the tumor-immune environment to attack remaining glioma cells and significantly minimized post-operative GBM recurrence in a mouse model [73].

Long-acting dosage forms must be biocompatible, non-toxic, non- or minimal swelling, and pathogen-free. Due to the unique physiological environment in the brain, biodegradable materials that are biocompatible when administrated systemically do not necessarily have good biocompatibility in the CNS. A previous study showed that poly(methylidene malonate 2.1.2.) implants were biocompatible via IV administration [82], whereas they had a strong neuroinflammatory response due to acidic degradants of the polymer [83]. Moreover, sterilization techniques such as ethylene oxide (EtO) may affect release characteristics of therapeutics loaded in a polymeric implant, thus resulting in shorter drug release durations and/or reduced amount of payload release [84]. Although protein immobilization in the polymeric matrix was used to adjust payload release rate, incomplete payload release was observed, which may have resulted from the polymer-protein interaction during release [74].

### 3.3. Nanotechnology-Based Delivery Systems

Nanotechnology-based therapeutics have demonstrated clinical successes in the treatment of various diseases such as cancer [85] and COVID-19 [86]. Liposomes and polymeric nanoparticles (NP) are the most widely studied nanocarriers for delivering peptides and proteins in the CNS (Table 2 and Table 3). In addition, lipidoid telodendrimer NP, block ionomer complexes, and nanostructured lipid NP have been investigated for CNS delivery of peptides and proteins. Peptides/proteins can either be encapsulated into NP [76,79] or form a nanocomplex with lipids or polymers via electrostatic interaction [78,80,87]. Schematic diagrams of these nanoparticulate carriers were shown in Figure 3. The nanocarriers are often incorporated into a long-acting dosage form (e.g., hydrogel) to provide sustained payload release and combinational therapy.

It has been shown that NP properties (e.g., size, shape, charge, and surface properties) are critical to their distribution in the CNS. NP with a small size may have drug loading limitations, whereas NP with a size larger than 200 nm may have poor transport in the brain and hence insufficient brain accumulation. The shape of NP can modulate cell adhesion and uptake, which in turn affects NP transport in the brain. On the other hand, the surface charge of NP can affect its transport within the target tissue [93]. Donaghue et al., encapsulated NT-3 into PLGA NP (50/50, 7–17 kDa, acid-terminated) with a particle size of 220 nm. IT administration of this NT-3 nano-therapeutic achieved functional improvement in rats with injured spinal cords [88]. Wang et al., created lipidoid-telodendrimer binary hybrid NP to deliver a protein therapeutic DT390, a modified diphtheria toxin with nonspecific cytotoxicity eliminated. CED administration of this hybrid NP significantly inhibited brain tumor growth in mice [89]. Moreover, Zhang et al., used lectin-modified polyethylene glycol-PLGA (PEG-PLGA) NP to encapsulate basic fibroblast growth factor (bFGF) against AD. IN delivery of this nanoparticulate system significantly improved the spatial learning and memory of AD rats compared to rats following IV administration of the same treatment [90]. In another study, Zhao et al., designed a new phospholipid-based gelatin NP to encapsulate bFGF against PD, and IN administration of this system showed obvious therapeutic effects (e.g., stimulation of the dopaminergic function in surviving neurons and partial conversion of the loss of dopaminergic neurons) in PD rats [91]. These studies confirmed the effectiveness of IN delivery of nanotechnology-based growth factors for the treatment of neurogenerative diseases.

### 3.4. Extracellular Vesicles

Extracellular vesicles (EVs) are small sacs enclosed by lipids that are released by cells and found in the space outside the cells. EVs themselves have been used for the treatment of CNS diseases and showed good safety. Intraparenchymal administration of exosomes extracted from allogenic placenta mesenchymal stem cells showed good safety in malignant middle cerebral artery infarct patients in a pilot randomized clinical trial [94]. In an ongoing clinical trial, safety and efficacy of EVs derived from allogenic adipose mesenchymal stem cells through IN administration are being tested for the treatment of mild to moderate dementia [95]. Mohammed et al., utilized EVs from primary rodent neural stem cells (NSC) to treat spinal cord injuries. IT injection of these EVs improved motor function and neuroprotection and reduced lesion volume and expression of inflammasome proteins in a rat model [96].

EVs have also been extracted and used for local delivery of protein therapeutics. In animal studies, Haney et al., found that neuronal cells readily internalized exosomes, and a significant number of exosomes were observed in the brains of PD mice after IN administration. Encapsulated protein-catalase exhibited significant neuroprotective effects in both in vitro and in vivo PD models [97]. Izadpanah et al., demonstrated that IN administration of EVs containing neprilysin, an enzyme that clears normal aggregated β-amyloid (Aβ) sheets in the brain, ameliorated brain-related behavioral function in an AD rat model [57]. Hayes et al., developed a modified, peroxisome-targeting, catalase derivative, enzyme-catalase-SKL, which was loaded into EVs through a co-incubation procedure, and the system was tested against multiple neurodegenerative diseases such as AD and PD. IN administration of catalase-SKL-loaded EVs achieved successful catalase-SKL delivery into the brain and did not show any off-target toxicity in healthy mice [98].

### 3.5. Live-Cell Therapy

Live cells represent the next generation of therapeutics since the cells can be engineered to secrete peptide and protein therapeutics, serving as drug depots (Figure 4) [99]. Genetically engineered therapeutic-secreting depots can mimic natural pulsatile or stimuli-responsive drug release behavior without the risk of fragile biologics losing their bioactivity during the release and transport process in vivo [100]. Currently, there are multiple clinical studies of chimeric antigen receptor (CAR) T cell therapies against GBM through direct intraparenchymal injection [101,102,103]. Portnow et al., produced an immortalized, clonal human NSC line to stably express cytosine deaminase, which can convert an oral prodrug, 5-fluorocytosine (5-FC), to its active form, 5-fluorouracil (5-FU), against GBM. The concentration of 5-FU increased locally in the human brain following ICV injection of NSC [104]. Killer et al., compared ICV delivery of mesenchymal stromal cells (MSC) that produced recombinant epidermal growth factor (rEGF) with a rEGF solution. Although the dose of this cell therapy needs to be further adjusted, the feasibility of using MSC-rEGF to preserve neurogenesis in the brain of irradiated mice was demonstrated [105]. Live cell treatment can also be achieved through IT administration or implantation. Aronson et al., showed that IT delivery of the live fibroblast therapy maintained a detectable level of fibroblasts that secret human erythropoietin in the CSF without any negative impact on inflammatory response in healthy rat brains [106]. In another study, Kauer et al., engineered a simulated extracellular matrix (sECM) composed of hyaluronic acid and PEG diacrylate-based hydrogel to encapsulate stem cells that released a tumor-selective tumor necrosis factor-related apoptosis-inducing ligand (TRAIL) in the tumor resection cavity over a period of 12 days, which eradicated residual tumor cells and significantly increased survival in a patient-derived xenograft GBM model [58].

## 4. Future Perspectives

Despite recent advances in drug delivery strategies, the BBB remains a physiological barrier that hinders peptide/protein distribution in the brain following systemic administration. Therefore, comprehensive approaches that address both delivery and stability challenges that peptide and protein therapeutics are facing are critically important. Although local delivery strategies can diametrically increase CNS concentration of protein and peptide therapeutics, complications from surgical implantation of devices (e.g., CED or IT injection port) and potential device failure due to foreign body responses will need to be fully understood and mitigated. Nanotechnology-based delivery systems administrated directly through local delivery strategies or incorporated into hydrogel for injection/implantation have been effective in CNS delivery of proteins and peptides. However, the nanoparticle fabrication process involves high shearing and stress conditions which may denature or degrade peptides and proteins. Therefore, it is critical to preserve the bioactivity of peptides and proteins during the nanoparticle fabrication process, and incorporating protectants (such as histidine, albumin) may be necessary. Moreover, polymer end groups (e.g., carboxyl group) may interact with peptide/protein interaction during manufacturing, release, and storage, resulting in altered and/or incomplete payload release [107]. Accordingly, understanding the polymer–biomolecule interaction is essential to minimize its impact on drug release of long-acting protein/peptide dosage forms. Importantly, suitable sterilization approaches and environmental controls need to be considered in order to ensure sterility and free-of-pyrogen of protein/peptide formulations for localized CNS delivery.

Cell-based therapies (e.g., EVs, live cells) have been emerging as the frontier of personalized medicines over the past few years. Most recently, an extracellular contractile injection system with a nanosized range has been engineered from a special bacterium, *Photorhabdus asymbiotica,* to deliver a variety of proteins in the CNS through ICV administration, including native toxin polypeptides (Pdp1 and Pnf), green fluorescent protein, Cre recombinase, zinc finger nuclease, and CRISPR-associated (Cas9) endonuclease [108]. Exosomes from human umbilical cord mesenchymal stem cells were able to prevent and reverse spinal nerve ligation-induced pain in rats through IT injection [109]. Exosomes loaded with therapeutical peptides and proteins through IT infusion may be expected to treat various CNS diseases in the future. Although cell-based therapies have demonstrated a huge potential against CNS diseases, scaling-up extraction and bioengineering processes of functionalized cell-based therapies (e.g., EVs, live cells) remains a challenge. With the extensive efforts devoted to advancing bioengineering and manufacturing processes, it is expected that these next generation therapies will have more clinical successes and broader applications against CNS diseases.

## Figures and Tables

**Figure 1 pharmaceuticals-16-00810-f001:**
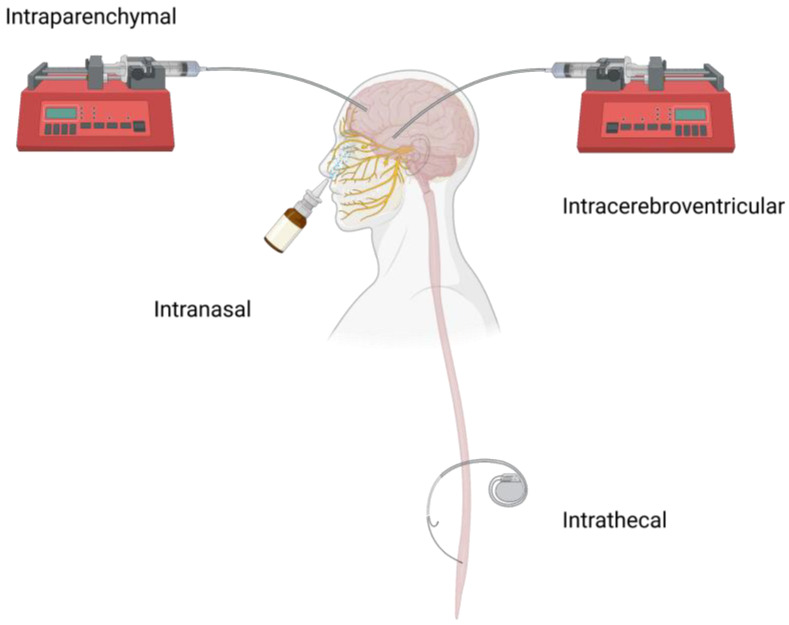
Illustration of the local CNS peptide and protein administration routes that bypass the BBB.

**Figure 2 pharmaceuticals-16-00810-f002:**
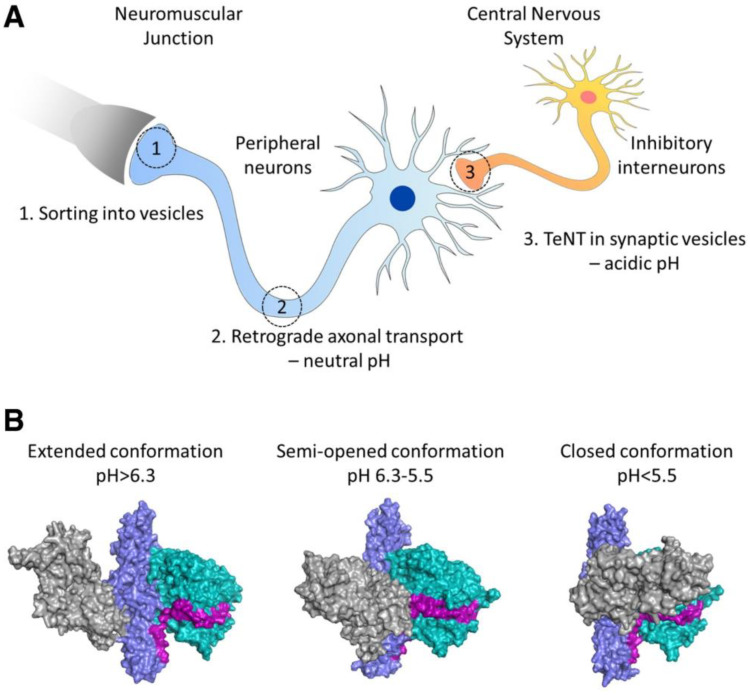
Illustration of the retrograde axonal transport of the tetanus toxin to CNS. TTC (TeNT) can move via axonal transport from the motor neurons of the neuromuscular junction to the motor neurons of the spinal cord and brainstem (**A**). This transport process is pH-dependent. pH mediated TTC domain rearrangements and transport between neurons (**B**). Reproduced with permission from [63].

**Figure 3 pharmaceuticals-16-00810-f003:**
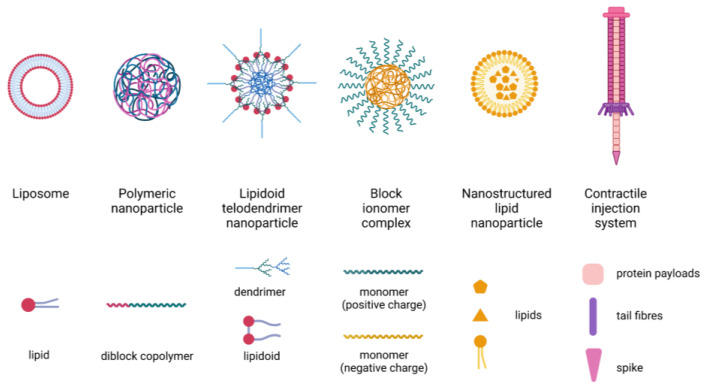
Schematic diagram of nanoparticulate carriers for CNS local delivery of peptides/proteins.

**Figure 4 pharmaceuticals-16-00810-f004:**
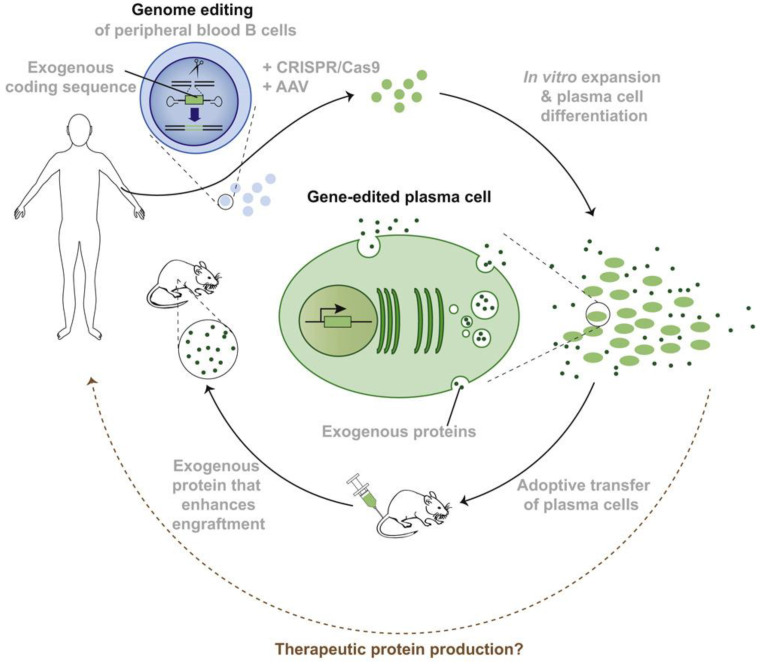
Illustration of genetically engineered therapeutical protein-secreting depots. Genome editing of cells isolated from patients under conditions of rapid cell expansion achieved targeted sequence integration and secretion of therapeutic polypeptides at high levels, and then adoptive transfer of these cells led to their engraftment in patients. Reproduced with permission from [99].

**Table 1 pharmaceuticals-16-00810-t001:** Pros and cons and recent clinical applications of main local CNS delivery strategies for proteins and peptides.

	Pros	Cons	Clinical Applications
ICV	Direct introduction of nearly 100% therapeutics into the CSF in the lateral ventricle with minimal drug-protein bounding.	Low drug distribution in the brain parenchyma that is away from the injection site; Bulk flow of CSF may result in fast clearance and systemic exposure; Complications associated with ICV device implantation.	Brineura [16]; tripeptidyl peptidase 1 (Phase I/II) [17,18]; idursulfase-β (Phase I/II) [19]; telbermin (Phase I) [20]
CED	Improved drug distribution in the brain parenchyma; Topographically more restricted, thus less complications compared to ICV.	Inefficient diffusion in the parenchyma for some macromolecules; Complications from the invasiveness of the CED.	Omburtamab (Phase I) [21,22]
IT	Direct injection of nearly 100% therapeutics into the CSF in the spinal subarachnoid space with long half-life in the CSF due to minimal drug-protein bounding and metabolism; Less invasive compared to ICV and CED	Less macromolecule distribution in the parenchyma compared to ICV and CED; Potential complications from IT device implantation.	Recombinant human arylsulfatase A(Phase I/II) [23]
IN	Noninvasive	Potential systemic exposure; Device design is critical.	Insulin & insulin detemir (Phase II) [6]; neuropeptide Y (Phase I) [7]

**Table 2 pharmaceuticals-16-00810-t002:** Representative long-acting formulations for protein and peptide delivery in the CNS.

Disease Model	Formulation	Cargo Therapeutic	Carrier Material	Release Duration	Ref.
GBM (GL261 mouse model)	Hydrogel	CXCL-10	Self-assembled oligopeptide	>12 days	[73]
Spinal cord injury (Rat model)	Hydrogel	basic fibroblast growth factor (bFGF) or FGF2	Heparin-modified poloxamer and lyophilized acellular spinal cord	>7 days	[74,75]
	Nanoparticle (NP)-hydrogel composite	neurotrophin-3 (NT-3) + antibody 11c7 (anti-NogoA)	Hyaluronan-methylcellulose (HAMC) and poly(lactic-co-glycolic acid) (PLGA)	NT-3: >58 days; anti-NogoA: >10 days	[76]
	Hydrogel	NT-3	Heparin-contained collagen	>90 days	[77]
	NP-hydrogel composite	chondroitinase ABC (ChABC) + stromal cell-derived factor 1α (SDF)	SH3 binding peptide-modified methylcellulose and PLGA	ChABC: >7 days; SDF: >14 days	[78]
	Liposome-hydrogel composite	brain-derived neurotrophic factor (BDNF) + acidic FGF (aFGF)	Heparin-modified poloxamer and scar-targeted tetrapeptide-modified liposomes	BDNF/aFGF: >21 days	[79]
	NP-hydrogel composite	BDNF	Agarose + polysaccharide polyelectrolyte complexes	>17 days	[80]
	Hydrogel	Anti-inflammatory peptide KAFAK + BDNF	HAMC	KAFAK/BDNF: >4 days (only ~50% of payloads released)	[81]

**Table 3 pharmaceuticals-16-00810-t003:** Representative nanotechnology-based protein and peptide delivery in the CNS.

Disease Model	Formulation	Cargo Therapeutic	Administration	Ref.
Spinal cord injury (Rat model)	PLGA NP	NT-3	IT	[88]
Brain tumor (Mouse model)	Lipidoid-telodendrimer binary hybrid NP	DT390	CED	[89]
AD (Rat model)	Lectin-modified PEG-PLGA NP	bFGF	IN	[90]
PD (Rat model)	Phospholipid-based gelatin NP	bFGF	IN	[91]
Mouse model	Chitosan coated nanostructured lipid carriers	human IGF-I (hIGF-I)	IN	[92]

## Data Availability

Not applicable.

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
