# Peer review of "Local Delivery Strategies for Peptides and Proteins into the CNS: Status Quo, Challenges, and Future Perspectives"

_pharmaceuticals, 2023, doi:10.3390/ph16060810_

Round 1

Reviewer 1 Report

In this review, the authors discussed the status quo, challenges, and future perspectives on local delivery of peptide and protein drugs into the CNS. The structure and writing ideas of the whole review are very clear. However, there are some aspects need to be improved.

1. The manuscript mainly expounds the Status Quo, Challenges, and Future PerspectivesLocal Delivery Strategies for Peptides and Proteins into the CNS. However, in the Future Perspectives section, the author should analyze and predict multiple solutions to solve the key problem of "blood-brain barrier", rather than just Cell therapy.

2. The abbreviation that appears for the first time in the manuscript should negotiate the full name, such as BBB in line 45.

Minor editing of English language required

Author Response

  1. In this review, the authors discussed the status quo, challenges, and future perspectives on local delivery of peptide and protein drugs into the CNS. The structure and writing ideas of the whole review are very clear. However, there are some aspects need to be improved.

    Response: Thank you for your time and positive feedback.

    1. The manuscript mainly expounds the Status Quo, Challenges, and Future Perspectives Local Delivery Strategies for Peptides and Proteins into the CNS. However, in the Future Perspectives section, the author should analyze and predict multiple solutions to solve the key problem of "blood-brain barrier", rather than just Cell therapy.

    Response: Thank you for the comment. The manuscript is mainly focused on local delivery strategies that can overcome the blood-brain barrier. We have provided few future perspectives on various local delivery strategies (please refer to section 4, paragraph 1 for details).

    1. The abbreviation that appears for the first time in the manuscript should negotiate the full name, such as BBB in line 45.

    Response: Thank you for the comment. We have revised the manuscript accordingly.

Reviewer 2 Report

Comments :

Abstract Last Sentence: Two much lengthy, not making a complete sense regarding the paper manuscript. Make it simple and specific.

Figure 1: Elaborate the routes as more routes are available.

Section 2 Local Delivery: Reference 15 written again and again, was required one time at the end of the paragraph.

Section 2.1, 2.2 and 2.3: Could have better explained in tabular form and their comparative study will display a clear picture of the various routes

Section 3.1: This section needs to put more available advanced studies.

Section 3.3: Clinical role of nanotechnology in covid-19 is not justified, needs an accurate reference.

Nanoparticles correlation with BBB regarding surface charge properties is missing.

Future perspectives: Here some problems were discussed while the solution to the problem was not discussed. It needs some rectification.

minor language and sentence making editing is required throughout the manuscript.

Author Response

Abstract Last Sentence: Two much lengthy, not making a complete sense regarding the paper manuscript. Make it simple and specific.

Response: Thank you for your comment. The last sentence of the abstract only has 10 words. We rephrased the abstract to make it concise.

Figure 1: Elaborate the routes as more routes are available.

Response: This paper is mainly focused on the local administration routes that can bypass the blood brain barrier (BBB). Therefore, we didn’t include other administration routes.

Section 2 Local Delivery: Reference 15 written again and again, was required one time at the end of the paragraph.

Response: Thank you for this comment. We have revised the manuscript accordingly.

Section 2.1, 2.2 and 2.3: Could have better explained in tabular form and their comparative study will display a clear picture of the various routes

Response: Please refer to Table 1 in our original submission. It summarizes the comparative study of various administration routes.

Section 3.1: This section needs to put more available advanced studies.

Response: Thank you for your comment. We highlighted a few studies that were through local CNS delivery routes. There are other advanced studies available. However, many of them were not through local CNS administration routes, which is the focus of the manuscript.

Section 3.3: Clinical role of nanotechnology in covid-19 is not justified, needs an accurate reference.

Nanoparticles correlation with BBB regarding surface charge properties is missing.

Response: We have cited references as suggested (references 85 and 86). We have also included the surface charge impact in the manuscript.

Future perspectives: Here some problems were discussed while the solution to the problem was not discussed. It needs some rectification.

Response: Thank you for your comment. We have provided perspectives on how to address some of the issues. For example, utilizing protectants (such as histidine, albumin) to help preserve bioactivity of peptides and proteins during the nanoparticle fabrication process. We also emphasized that polymer-biologic interaction needs to be thoroughly studied to ensure the stability and well-controlled release characteristics of peptides and proteins.

Reviewer 3 Report

The manuscript discusses the potential applications of peptide and proteins in the treatment of human diseases, particularly their efficiency, role and challenges to the all-important CNS due to the presence of BBB. The authors also discuss various formulations strategies being employed and importantly presented future perspectives for efficient success in the field of CNS delivery. The manuscript is well written and organized.

Some suggestions should be addressed before the acceptance of the manuscript;

1-     The first couple of sentences should be removed as this information are available. (line 28-32).

2-     Although the authors have acknowledged Biorender.com, but the authors should provide permission letter to avoid copyrights. They usually provide.

3-     References should be manually corrected as the references are not uniform.

Although the topic and content are solid and fully addressed an important issue (CNS). Since CNS delivery is challenging, therefore every aspect should be addressed. In the context of CNS delivery, Exosome can play a crucial role, I would be glad to hear from the authors regarding the role of exosomes in CNS delivery as they can cross BBB. It is up to the authors whether they want to include exosomes or not, but to be honest it will increase readership.

Author Response

1-     The first couple of sentences should be removed as this information are available. (line 28-32).

Response: Thank you for your comment. We believe the content can increase readership to who is new to the area, so we decide to keep it.

2-     Although the authors have acknowledged Biorender.com, but the authors should provide permission letter to avoid copyrights. They usually provide.

Response: Thank you for your comment. They have already been included in the “permission.zip” file.

3-     References should be manually corrected as the references are not uniform.

Response: Thank you for your comment. References have been checked and modified.

Although the topic and content are solid and fully addressed an important issue (CNS). Since CNS delivery is challenging, therefore every aspect should be addressed. In the context of CNS delivery, Exosome can play a crucial role, I would be glad to hear from the authors regarding the role of exosomes in CNS delivery as they can cross BBB. It is up to the authors whether they want to include exosomes or not, but to be honest it will increase readership.

Response: Thank you for your comment. The content for exosome has already been added into the manuscript.